TOPICAL REVIEW

# The complementary dominance hypothesis: a model for remediating the 'good' hand in stroke survivors

Nick M. Kitchen[1,2], Brooke Dexheimer[3] , Jisung Yuk[2] , Candice Maenza[1], Paul R. Ruelos[2], Taewon Kim[2,4,5] and Robert L. Sainburg[1,2,5]

[1] *Department of Neurology, College of Medicine, Pennsylvania State University, Hershey, Pennsylvania, USA*
[2] *Department of Kinesiology, Pennsylvania State University, University Park, Pennsylvania, USA*
[3] *Department of Occupational Therapy, Virginia Commonwealth University, Richmond, Virginia, USA*
[4] *Department of Physical Medicine and Rehabilitation, College of Medicine, Pennsylvania State University, Hershey, Pennsylvania, USA*
[5] *Huck Institute of the Life Sciences, Pennsylvania State University, University Park, Pennsylvania, USA*

Handling Editors: Laura Bennet & Richard Carson

The peer review history is available in the Supporting Information section of this article (https://doi.org/10.1113/JP285561#support-information-section).

**Abstract** The complementary dominance hypothesis is a novel model of motor lateralization substantiated by decades of research examining interlimb differences in the control of upper

N. M. Kitchen and B. Dexheimer contributed equally to this work.

extremity movements in neurotypical adults and hemisphere-specific motor deficits in stroke survivors. In contrast to earlier ideas that attribute handedness to the specialization of one hemisphere, our model proposes complementary motor control specializations in each hemisphere. The dominant hemisphere mediates optimal control of limb dynamics as required for smooth and efficient movements, whereas the non-dominant hemisphere mediates impedance control, important for countering unexpected mechanical conditions and achieving steady-state limb positions. Importantly, this model proposes that each hemisphere contributes its specialization to both arms (though with greater influence from either arm's contralateral hemisphere) and thus predicts that lesions to one hemisphere should produce hemisphere-specific motor deficits in not only the contralesional arm, but also the *ipsilesional* arm of stroke survivors – a powerful prediction now supported by a growing body of evidence. Such ipsilesional arm motor deficits vary with contralesional arm impairment, and thus individuals with little to no functional use of the contralesional arm experience both the greatest impairments in the ipsilesional arm, as well as the greatest reliance on it to serve as the main or sole manipulator for activities of daily living. Accordingly, we have proposed and tested a novel intervention that reduces hemisphere-specific ipsilesional arm deficits and thereby improves functional independence in stroke survivors with severe contralesional impairment.

(Received 3 February 2024; accepted after revision 25 April 2024; first published online 11 May 2024)
**Corresponding author** N. M. Kitchen: Department of Neurology, Penn State Health Milton S. Hershey Medical Centre, 500 University Dr., Hershey, PA, 17033, USA.    Email: nickkitchen1@gmail.com

**Abstract figure legend** Summary of the complementary dominance hypothesis of motor lateralization and its application in remediating functional motor deficits of the ipsilesional arm in stroke survivors. The complementary dominance model of motor lateralization highlights distinct yet complementary functional contributions of each hemisphere to movement control (top panel). The dominant (left) hemisphere specializes in optimal control of limb dynamics and is therefore advantaged for well-established behavioural patterns under predictable circumstances. The non-dominant (right) hemisphere specializes in impedance control, which regulates steady-state limb position and responses to unexpected stimuli in the environment. Unilateral stroke results in hemisphere-specific motor deficits in both the contralesional and ipsilesional arms that are functionally limiting and consistent with the complementary dominance hypothesis. Left hemisphere stroke leads to deficits in initial direction accuracy, resulting in curved hand trajectories (middle panel, right). By contrast, right hemisphere stroke impairs final position accuracy, as demonstrated by larger deviations from the target location (middle panel, left). Based on these observations, we are now testing a novel training intervention to remediate functional motor deficits of the ipsilesional arm for chronic, severely impaired stroke survivors. The intervention includes hemisphere-specific virtual-reality training (bottom panel, top row) and real-world dexterity training (bottom panel, lower row), which we expect to translate to improved functional independence.

## Neural lateralization: an overview

The mammalian brain is anatomically organized into left and right sides that share grossly homologous regions, yet the division of neural processes between the two hemispheres is not symmetric for a large number of neuro-behaviors. Whereas primary and secondary motor and sensory regions are largely symmetric, tertiary processing

**Nick Kitchen** is a postdoctoral researcher at Pennsylvania State University in Dr Robert Sainburg's research group, where he is investigating the lateralized neural mechanisms underlying upper limb movement control and motor learning in health and neurological injury. He obtained his PhD from the School of Psychology at the University of Birmingham (UK) in 2018 and trained previously as a postdoctoral researcher in the Department of Speech and Hearing Sciences at the University of Washington. **Brooke Dexheimer** is an assistant professor and occupational therapist at Virginia Commonwealth University, conducting basic and translational research on underlying mechanisms of motor control and motor deficits in individuals with neurological diseases. She completed her PhD in Kinesiology and Graduate Certificate in Translational Science from Pennsylvania State University. She is also an NIH National Centre for Advancing Translational Sciences (NCATS) Fellow. Prior to her PhD, she earned a doctorate in Occupational Therapy from Washington University in St. Louis.

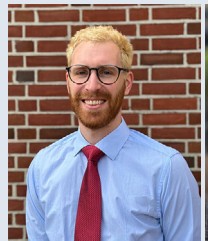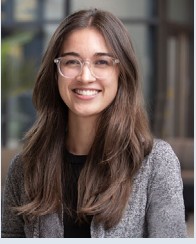

areas that do not receive or send information to and from the periphery tend to be asymmetric. This tendency for some neural processes to be specialized to one hemisphere is referred to as brain lateralization and is evident across a range of vertebrate species (for reviews see Bisazza et al., 1998; Vallortigara & Rogers, 2005). Studies of animal behaviour, for example, have shown directionally biased responses to threatening predator stimuli (Cantalupo et al., 1995; Rogers, 2000), as well as for more routine behaviours like searching for food and eating (Clapham et al., 1995; Mench & Andrew, 1986). Lateralization is easy to understand from a neural conservation perspective: the least number of neurons required in a neurobehavioural circuit are contained within a single hemisphere and not duplicated between hemispheres. Thus, while in each hemisphere a specific region contains the intrahemispheric connections that can support a general function like language, the circuitry for specific aspects of this function may be located in either hemisphere. Indeed, in most humans the left hemisphere is dominant for processing lexicon and syntax for language and speech production (Geschwind, 1970; Milner et al., 1964; Wernicke, 1874), whereas the right hemisphere mediates prosody and emotional content of language (Borod et al., 1998; Bottini et al., 1994).

Lateralization in the motor system is reflected, in part, by handedness, a phenomenon in which the control of one arm and hand appear better adapted for specific aspects of some tasks or components of tasks. In contrast to the language system, which has one set of midline articulators, the motor system of the upper extremities has two independent articulators (the left and right arms) that, from a sensorimotor perspective, are connected to their respective contralateral hemispheres. Thus, the upper limb motor system is unique in having the opportunity for asymmetry in behaviour, as well as neural function. Handedness is heavily biased to the right side, with ~90% of the population estimated to be right-handed (Coren & Porac, 1977; Gilbert & Wysocki, 1992). In line with this observation, early theories of motor lateralization suggested that the dominant, left hemisphere assumes a major role in controlling movement, whereas the non-dominant, right hemisphere assumes a minor role by relying on motor commands sent from the major hemisphere via the corpus callosum (Annett et al., 1979; Liepmann, 1905). This view was supported by studies showing that, for right-handers in particular, activation of motor areas in the cerebral hemisphere ipsilateral to the dominant arm is lower than ipsilateral activation measured during non-dominant arm movements (Dassonville et al., 1997; Kansaku et al., 2005; Kawashima et al., 1997; Kim et al., 1993; Taniguchi et al., 1998; Viviani et al., 1998). Although this does not account for the fact that ipsilateral cortex activity is still present during unimanual, dominant arm movements

(Dassonville et al., 1997; Kansaku et al., 2005; Taniguchi et al., 1998) and that ipsilateral cortex activation has been shown to encode specific features of limb movement (Ganguly et al., 2009), the contribution of ipsilateral cortices is still often overlooked in studies of upper limb movement control, suggesting some continued belief that each hemisphere only contributes control to the contralateral arm (Fig. 1*A*). Such global dominance hypotheses (Dexheimer & Sainburg, 2021; Jayasinghe et al., 2020; Woytowicz et al., 2018) suggest that the dominant arm should show better performance than the non-dominant arm on all aspects of motor function (Annett et al., 1979; Volkmann et al., 1998; Ziemann & Hallett, 2001). However, substantial research has now demonstrated that specific tasks or features of tasks are associated with better performance by the non-dominant arm (Bagesteiro & Sainburg, 2003; Dexheimer & Sainburg, 2021; Duff & Sainburg, 2007; Yadav & Mutha, 2020) and hemisphere-specific components of movement are selectively impaired in both arms as a result of left or right hemisphere damage following a stroke (Mani et al., 2013; Mutha et al., 2011a, 2011b; Schaefer et al., 2009a, 2009b, 2012; Winstein & Pohl, 1995). These findings, among others, lend support to an alternative model of motor lateralization, which we previously named the dynamic dominance hypothesis, but hereafter refer to as the *complementary dominance hypothesis*. This model proposes different specialized, yet complementary, roles of the left and right hemispheres for motor control (Sainburg, 2002, 2005). According to this hypothesis,

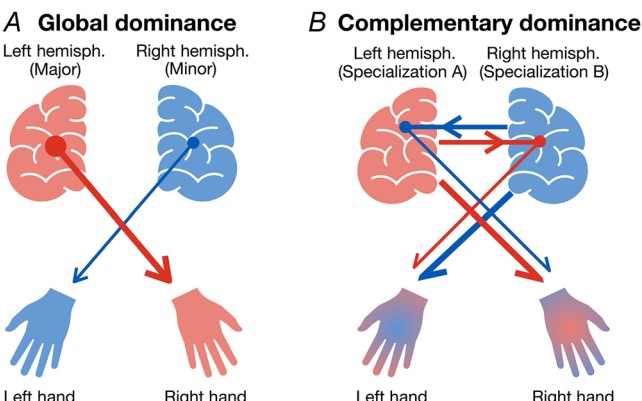

**Figure 1. Two hypotheses of motor lateralization**
*A*, the global dominance hypotheses posit that the dominant right arm should exhibit superior performance on all aspects of motor function (thick red arrow) compared to the non-dominant left arm (thin blue arrow), with influence of the ipsilateral cortices commonly overlooked. *B*, the complementary dominance hypothesis posits that each hemisphere has different, but complementary, specialized functional contributions to movement control. Under this hypothesis, *both* hemispheres contribute to control of unilateral arm movements, though with a greater influence of the specialized function of the contralateral hemisphere.

for right-handers, the dominant left hemisphere is specialized for optimal control of limb dynamics, and tends to result in smooth and mechanically efficient movement trajectories for the dominant arm. Conversely, the non-dominant, right hemisphere behaves more like an impedance controller that results in better stabilization of the left arm under unpredictable task or environmental conditions, and more stable achievement of steady-state final positions of reaching movements. For left-handers, the same specializations should occur, but in the opposite hemisphere and arms (Przybyla et al., 2011). Importantly, this hypothesis of motor lateralization outlines that *each* hemisphere contributes to control of both arms, but with greater influence of the specialized function from the contralateral hemisphere. Figure 1*B* shows this arrangement, in which each hemisphere has a major influence on its contralateral arm, and a minor but important influence on the ipsilateral arm.

## Motor lateralization

**Lateralized motor behaviour.** During multi-joint reaching movements, it is necessary to generate muscle forces that account for both external forces imposed by the environment, such as gravity and object inter-action forces, as well as forces arising from within the body itself, such as inertial interactions between limb segments and soft tissue viscoelasticity (Sainburg & Kalakanis, 2000). Coordination in the extent and timing of muscle torque generation at the elbow and shoulder joints must account for these factors in order to produce intended trajectories towards the target. Investigations of intersegmental dynamics in healthy,

neurologically intact participants have demonstrated that the dominant, right arm produces smoother and more efficient trajectories by coordinating elbow and shoulder torques more effectively than the non-dominant, left arm (Bagesteiro & Sainburg, 2002; Sainburg, 2002; Sainburg & Kalakanis, 2000; Schaffer & Sainburg, 2017; Tomlinson & Sainburg, 2012). Moreover, trajectory deviations of the non-dominant arm increase as the required movement involves a greater influence of intersegmental dynamics, while the dominant arm curvature remains similar across all movements (Sainburg & Kalakanis, 2000). This is exemplified by the dominant and non-dominant hand and arm trajectories of a representative participant in Fig. 2*A*, which show that the dominant arm produces smoother trajectories with smaller deviations when reaching towards a medially located target. The group mean elbow torque profiles in Fig. 2*B* demonstrate that the dominant arm (right panel) achieves this with minimal muscle torque (long-dash line), exploiting the inter-action torque (short-dash line) from shoulder motion that drives the elbow into extension. Conversely, notably larger muscle torque is generated by the non-dominant arm (Fig. 2*C*, left panel), which fails to predictively integrate interaction torque from shoulder motion in a similar way to the dominant arm. This results in more energetically efficient movements of the dominant arm (Bagesteiro & Sainburg, 2002; Sainburg, 2002; Sainburg & Kalakanis, 2000), that are apparent in Fig. 2*C* from the lower group mean shoulder and elbow joint torque impulses of the dominant arm (hashed bars) than non-dominant arm (grey bars) for rapid, multi-joint reaching movements. Accordingly, the dominant arm controller appears well adapted to take advantage of limb mechanics when planning and controlling movements.

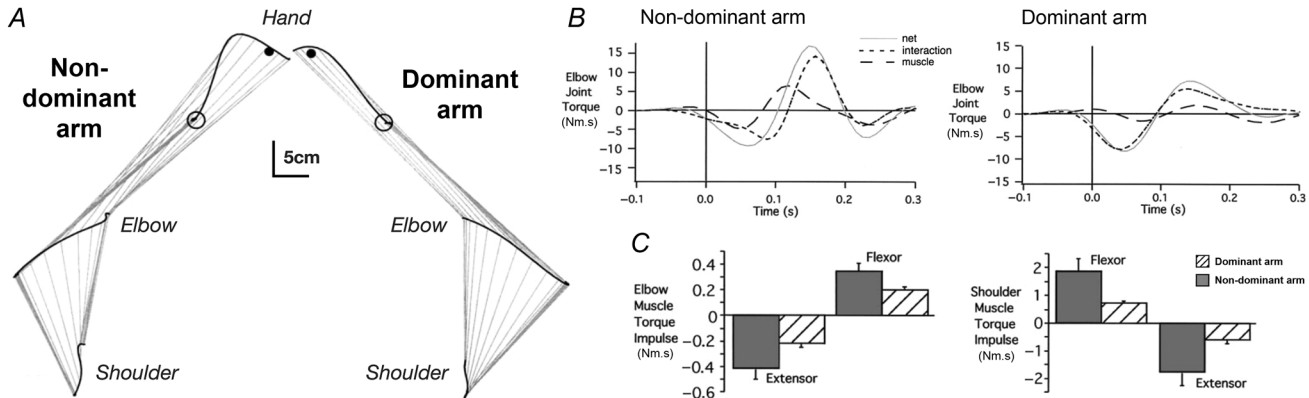

**Figure 2. Evidence of a dominant arm advantage in accounting for intersegmental dynamics during rapid, multi-joint reaching**
*A*, trajectories of the shoulder, elbow and hand of the non-dominant (left trace) and dominant (right trace) arms for one representative participant during reaches to a medially located target. *B*, group mean elbow joint torque profiles of the non-dominant (left panel) and dominant (right panel) arms. *C*, group mean (with 1 standard error bars) elbow (left panel) and shoulder (right panel) integrated muscle torque impulses for reaching movements of the non-dominant (grey bars) and dominant (hashed line bars).
Figure modified from Bagesteiro and Sainburg (2002).

Similar dominant arm advantages have been reported for reaching movements that include the additional dynamic effects of torso rotation on the arm (Pigeon et al., 2013), as well as during skilled overarm throwing performance (Hore et al., 1996, 2005). Collectively, these results suggest the dominant arm may be specialized for predicting and accounting for the impending mechanics of movements more accurately than the non-dominant arm.

In contrast to this specialization of dominant arm motor control, the non-dominant arm appears to play a complementary role in the control of voluntary arm movements. Whereas the dominant arm controller seems to better predict the impending dynamics of movements, the non-dominant arm shows advantages in controlling impedance for stabilization of limb position and trajectory in the face of unexpected perturbations (Sainburg, 2002; Schabowsky et al., 2007). While this can result in lower final position errors of the non-dominant arm than the dominant arm during unperturbed movements (Sainburg & Kalakanis, 2000; Wang & Sainburg, 2007), the specialized role of the non-dominant arm as an impedance controller is most apparent from studies that imposed external loads or forces to the arm during movement. For example, when unpredictable and inconsistent dynamic perturbations are applied to the arm during single-joint (Bagesteiro & Sainburg, 2003) and multi-joint (Yadav & Sainburg, 2014a) arm movements, the non-dominant arm opposes these forces better than the dominant arm. This is apparent in Fig. 3A, showing that mean

squared jerk values of reaching movements made in an inconsistent force environment were lower – and hence smoother – for the non-dominant arm (shaded diamonds, solid lines) than the dominant arm (grey squares, dashed lines). However, when a constant or more predictable force perturbation is applied over repeated trials, the dominant arm is advantaged (Duff & Sainburg, 2007; Yadav & Sainburg, 2014a), although online movement correction and final position accuracy still appear to be more proficient in the non-dominant limb (Duff & Sainburg, 2007; Schabowsky et al., 2007; cf. Yadav & Sainburg, 2014a). This is demonstrated in Fig. 3B (same symbol and colour key as outlined above for Fig. 3A), indicating that the dominant arm makes smoother reaches (lower mean squared jerk) than the non-dominant arm when moving in a consistent, predictable force environment. Moreover, during conditions of more predictable force perturbations, the dominant arm tends to show greater persistence of load-adapted behaviour when the perturbation is removed, or task conditions are otherwise changed (Duff & Sainburg, 2007; Schabowsky et al., 2007). Collectively, this implies the dominant arm controller forms more robust models of the task and environmental conditions that can be used to control movement in a predictive manner. In contrast, the non-dominant arm controller governs limb impedance to better compensate for unpredictable or unfamiliar task conditions (e.g. Dexheimer & Sainburg, 2021; Yadav & Mutha, 2020). The specialization for impedance control in

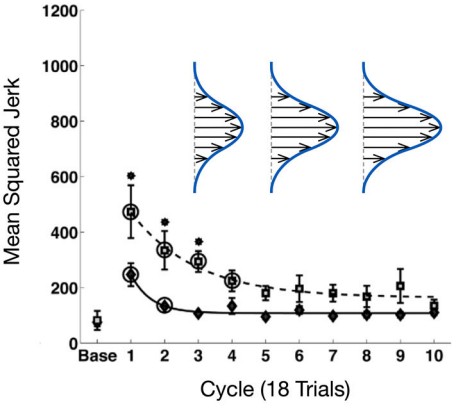
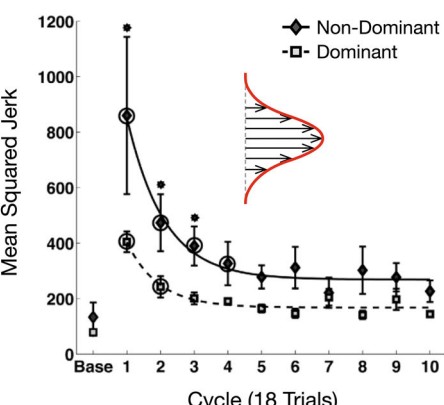

**Figure 3. Limb-specific reaching performance advantages under predictable and unpredictable dynamic environments**

Mean squared jerk of reaching movements performed under two types of novel force-fields that perturbed movements in a direction perpendicular to reaching direction: *A*, an inconsistent field which changed strength between trials and varied linearly with movement velocity (blue, inset); and *B*, a consistent field of a fixed strength that varied with the square of velocity (red, inset). Data are presented as group means (with 1 standard error bars) for binned cycles of 18 trials (total 180 trials). Non-dominant arm movements are shown as shaded diamonds, solid lines and dominant arm movements are shown as grey squares, dashed lines. Star symbols indicate cycles that are significantly different between arms, and circles indicate cycles that were significantly different from the final cycle within an arm.

Figure modified from Yadav and Sainburg (2014a).

the non-dominant arm may, in part, be achieved by greater reliance on proprioceptive feedback, which is supported by observations of enhanced limb position and motion matching in the non-dominant arm (Goble & Brown, 2009, 2010; Goble et al., 2009), as well as increased right hemisphere activity during kinesthetic tendon vibration applied to both the left and right wrist (Naito et al., 2005). However, modulation of stretch reflex gains appears to be *reduced* in the non-dominant arm when preparing to make a goal-directed movement (Torell et al., 2023), and there is mixed evidence as to whether postural stretch reflex sensitivity is heightened in the non-dominant arm (Maurus et al., 2021; Walker & Perreault, 2015). Consequently, the non-dominant hemisphere may be specialized for control of limb impedance, at least in part, through proprioceptive feedback loops. However, recent findings in proprioceptive deafferentation have indicated that this specialization of the non-dominant arm for impedance control is also dependent on feedforward, open-loop control mechanisms (Jayasinghe et al., 2020).

It is important to note that the findings outlined here characterizing lateralized motor behaviours of the dominant and non-dominant arm have been performed predominantly on neurotypical, right-handed participants. Left-handed participants account for around 10% of the population (Coren & Porac, 1977; Gilbert & Wysocki, 1992) and are relatively heterogeneous in terms of their behaviour, tending to use their non-dominant arm more frequently for daily activities than right-handed individuals, according to data from handedness inventories (Oldfield, 1971; Przybyla et al., 2011). Although the true extent of handedness may differ from those measured by self-report methods (Raczkowski et al., 1974; Satz et al., 1967), this suggests that the patterns of hand-use in left-handers may not be a behavioural mirror image of right-handers. Nevertheless, in a study comparing dominant and non-dominant arm reaching of left- and right-handed participants, both groups of participants made reaches with their dominant arm that were directed relatively straight towards the target with similar final position accuracy, regardless of interaction torque amplitude that varied with target location (Przybyla et al., 2011). Conversely, non-dominant arm movements became more curved, with reduced initial direction accuracy, as interaction torque amplitude increased. However, this asymmetry was more pronounced in the right-handed than left-handed participants. In addition to more bilaterally distributed hand use in left-handers, this greater symmetry between control of the right and left arms may stem from greater ipsilateral cortex recruitment during non-dominant arm movements in left- than right-handers (Kawashima et al., 1997; Solodkin et al., 2001), but this issue remains unresolved (Dassonville et al., 1997; Kim et al., 1993; Li et al., 1996).

Considering that the studies discussed here only address unilateral limb movement, a natural follow-up question is whether similar patterns of lateralized motor behaviours are also evident during bimanual motor performance. Woytowicz et al. (2018, 2020) investigated this with a bimanual task intended to simulate the naturalistic action of slicing a loaf of bread, where one arm maintained its static position at a start location (akin to stabilizing the bread) as the other arm made out and back reaching movements (resembling slicing the bread with a knife). Since the two arms were mechanically linked by a spring, the reaching arm had to account for the spring forces to make smooth, accurate movements, while the stabilizing arm had to resist the changing spring forces to maintain its position. A diagram of this task is shown in Fig. 4*A* for the condition in which the non-dominant, left arm stabilized at the start position and the dominant, right arm reached toward a target and back. Importantly, the cursors were visually displaced from the location of the unseen hands, such that both cursors could be placed at the start location simultaneously without the hands physically colliding. Arm segment trajectories for one representative participant performing this task are shown in Fig. 4*B* and *C*. For left arm reaches (Fig. 4*B*), the hand trajectory deviated from the linear path to the target to create a curved trajectory. During these left arm reaches, the right hand showed large displacements from the position it was required to stabilize around. Conversely, for right arm reaches (Fig. 4*C*), hand trajectories showed only small deviations and were directed fairly straight towards the target. During this right arm reaching, the left arm position remained fairly stable. Group-level analysis reflected these initial observations in that the right arm made reaches with straighter trajectories overall (i.e. smaller deviations from linearity; Fig. 4*D*), which appears to have been due to the right arm generating muscle torques that better compensate for the spring and interaction torques acting on the elbow (Fig. 4*E*) than the left arm (Fig. 4*F*). Furthermore, when the left arm was the stabilizing arm, it was displaced by a smaller distance from the start location than the right arm (Fig. 4*G*) demonstrating better stabilization performance overall. This was achieved by reducing measured endpoint compliance (inverse of stiffness) of the left arm (Fig. 4*H*). It should be noted that the maximum spring forces experienced by both hands were the same (Fig. 4*I*). Hence, during this bimanual task the dominant, right arm made straighter, more accurate reaching movements with improved compensation for the torque induced at the joints by the spring, whereas the non-dominant, left arm was found to better stabilize its position with smaller displacements from the start location (Woytowicz et al., 2018, 2020). Although the interlimb differences in stabilizing hand displacement were statistically significant, it is worth noting that the

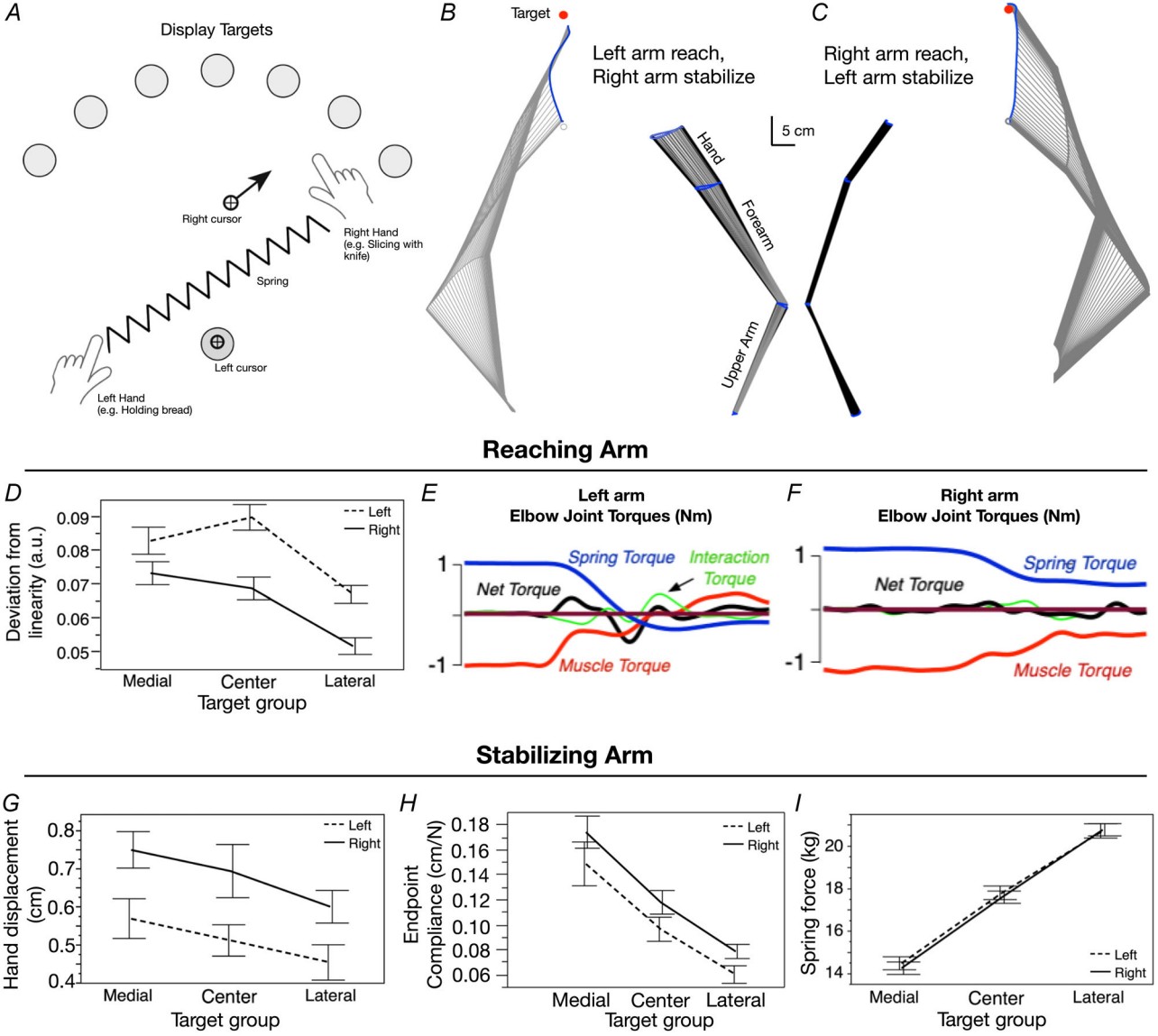

**Figure 4. Complementary advantages of dominant and non-dominant arm motor control in a functionally realistic bimanual task**

*A*, one hand (in this example the left) was required to maintain its position at the start location while the other hand (in this example the right) made out and back reaching movements to different targets. The two arms were physically coupled by a spring connected to the hands and visual cursors of the left and right hands were visually displaced from their unseen positions to align at the start location. This task was intended to simulate the action of using both hands to hold and slice a loaf of bread. *B* and *C*, arm segments and trajectories of a representative subject reaching and stabilizing with either arm. *D*, group mean deviation from linearity (with 1 standard error bars; lower values indicate straighter movements, a.u. = arbitrary units) of the dominant, right arm (solid line) and non-dominant, left arm (dashed line) performing the reaching action across sets of targets. *E* and *F*, elbow joint torque components of the dominant, right arm (*E*) and non-dominant, left arm (*F*) when performing the reaching action. *G–I*, group mean (with 1 standard error bars) of measures for the dominant, right arm (solid lines) and non-dominant, left arm (dashed lines) performing the stabilizing action across sets of targets: stabilizing hand displacement (*G*), stabilizing arm endpoint compliance (inverse of stiffness; *H*) and maximum spring force experienced by the stabilizing hand (*I*).

Data and figure modified from Woytowicz et al. (2018).

overall magnitude of displacement was relatively small (i.e. <1 cm), regardless of which hand was used to stabilize position. Nevertheless, while interlimb asymmetries in bimanual task performance have been reported previously (Dounskaia et al., 2010; Sherwood, 1994), these studies by Woytowicz and colleagues demonstrated distinct control advantages of the dominant and non-dominant arms that are predicted by the complementary dominance hypothesis during a functionally realistic bimanual task.

**Hybrid control.** The studies discussed so far have explored lateralized motor behaviour by examining differences in motor performance between the dominant and non-dominant arm. Although the contralateral cortical hemisphere predominantly influences control of unilateral limb movements, small but concurrent activation of the ipsilateral cortex (Dassonville et al., 1997; Kawashima et al., 1997; Kim et al., 1993) suggests that both hemispheres are engaged in controlling movements of a single limb (see following section for extended discussion

with additional evidence from unilateral stroke survivors). Hence, the different movement coordination patterns observed between the dominant and non-dominant arms might be attributable to different contributions of each hemisphere. Building on prior computational models of dominant arm reaching (Scheidt & Ghez, 2007), Yadav and Sainburg (2011, 2014b) explored this idea by proposing a hybrid control scheme to delineate the contributions of the dominant and non-dominant hemisphere to unilateral reaching of either arm. Those simulations were initially driven by an optimal controller that minimized task errors and energetic costs (simulating dominant hemisphere control) and transitioned to an impedance controller later in the movement that specified stiffness at the intended final position, as well as viscosity at joints in the arm (simulating non-dominant hemisphere control). Some example trajectories from these simulations are shown in Fig. 5A, which show variations in the point in the movement at which the controller transitions from predictive to impedance control ('switch instant', red square). Notably, Fig. 5A demonstrates how

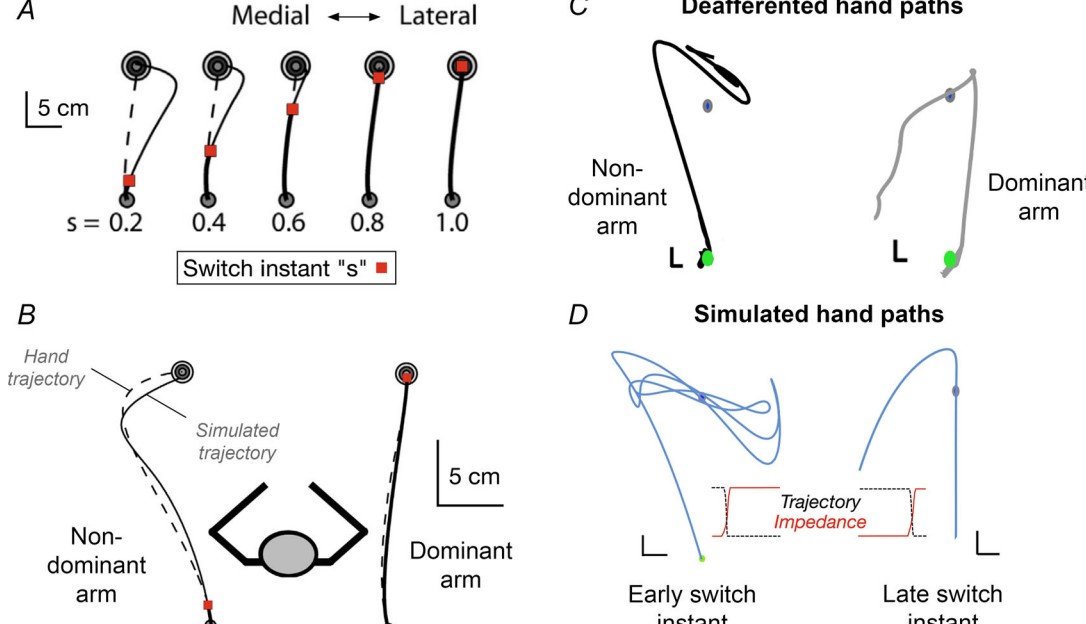

**Figure 5. Simulations of serially engaged trajectory and impedance controllers closely replicate dominant and non-dominant arm reaching in participants with and without somatosensation**
*A*, simulated hand paths (solid lines) with varying switch instants (red squares) from an initial trajectory controller to an impedance controller. *B*, simulated trajectories (solid lines) that most closely match recorded hand trajectories (dashed lines) of the non-dominant, left hand and dominant, right hand of a representative healthy (neurologically intact) participant. Switch instants of simulated hand trajectories are shown as red squares. *C*, non-dominant (left, black) and dominant (right, grey) hand trajectories of a chronically deafferented participant without visual feedback of hand position, and hence no sensory feedback of limb position (see text for details). Scale bars represent 2 cm. *D*, simulations of a deafferented participant using serially engaged trajectory (inset, black dash) and impedance (inset, red dash) controllers. Simulations that most similarly matched empirical hand trajectories show early (non-dominant arm) and late (dominant arm) switch instants from trajectory to impedance control. Simulations of reaches from both arms shown included a weak impedance controller that accounted for altered joint stiffness and damping due to deafferentation. Scale bars represent 2 cm.
*A* and *B*, modified from Yadav and Sainburg (2014b); *C* and *D*, modified from Jayasinghe et al. (2020).

the different switch points led to markedly different simulated hand trajectories. By altering the switch instant to optimally align the simulations with empirical reaching data, Yadav and Sainburg (2011, 2014b) found that dominant arm reaching was best characterized by a prolonged period of optimal control and a later transition (during the deceleration phase) to impedance control (Fig. 5*B*, right), while non-dominant reaching was more accurately represented by an earlier transition (prior to peak velocity) to impedance control (Fig. 5*B*, left). Thus, these simulations supported the idea that both hemispheres influence the control of the dominant and non-dominant arms in a complementary manner, but with a greater emphasis on the specialization of the contralateral hemisphere.

**Lateralization of open-loop control mechanisms.** Under these hybrid control simulations there is an assumption that accurate sensory information of limb position and velocity is available to both controllers. To address the question of how the integrated output of each controller might be affected by the absence of such sensory information, Jayasinghe et al. (2020) examined dominant and non-dominant arm reaching in an individual with a unique case of chronic proprioceptive deafferentation. This rare medical condition causes complete loss of large diameter sensory fibres in the periphery, but spares smaller diameter motor fibres, resulting in a total loss of somatosensation bilaterally in the trunk and extremities while retaining the ability to transmit motor commands to muscles (for detailed case reports see Cole & Paillard, 1995; for a recent review see Jayasinghe et al., 2021). Hence, experimentally occluding visual feedback of limb position for these individuals presents the unique ability for researchers to examine movements in the absence of sensory information about limb position or movement. Accordingly, when Jayasinghe et al. (2020) removed visual feedback of limb position, the proprioceptively deafferented participant (hereafter referred to by the initials GL) made reaches with notably different trajectories from age-matched controls. Whereas control participants reached with relatively straight trajectories and small terminal oscillations in both arms, GL showed profound interlimb differences in reaching performance, which can be seen in the representative hand path trajectories shown in Fig. 5*C*. Notably, the dominant, right arm was initially directed more accurately toward the target than non-dominant, left arm movements, but never stabilized at the end of movement. Instead, the movements slowly drifted away from the target following the initial motion near to the target. In contrast, the non-dominant left arm was less accurate than the dominant, right arm with respect to initial trajectory direction but oscillated near the target at the end of the movement. Thus,

the non-dominant arm showed worse trajectory control, but the ability to stabilize at the end of movement while the dominant arm showed better trajectory control, but poor ability to stabilize at the end of movement. A two-segment model of the arm with two, serially engaged open-loop controllers of optimized trajectory and postural impedance was simulated to operationalize these findings (Jayasinghe et al., 2020). After accounting for the altered joint stiffness and damping caused by deafferentation, there was a close resemblance between GL's non-dominant arm trajectories and the simulation that included a transition from the trajectory to the postural controller that occurred early in the movement (Fig. 5*D*, left). Conversely, GL's dominant arm trajectories showed a striking similarity to the simulation with a later transition to the postural controller (Fig. 5*D*, right). Notably, it was only the simulations that included a weak impedance controller (i.e. reflecting the altered joint stiffness and damping values due to deafferentation) which reflected the reaches GL made, including the distinctively large oscillations of the non-dominant arm (Fig. 5*D*, left) and position drift of the dominant arm (Fig. 5*D*, right) observed at the end of GL's movements (Fig. 5*C*). These observations suggest that reaching in the absence of somatosensory feedback can be described by hybrid control, but via two poorly tuned controllers. Moreover, the transition points between controllers of simulations that most closely matched GL's dominant and non-dominant arm reaches suggest a greater reliance on optimal trajectory control during dominant arm reaching, and a greater reliance on postural impedance control during non-dominant arm reaching (Jayasinghe et al., 2020). Accordingly, these findings extend previous work (Yadav & Sainburg, 2011, 2014b) to show that even in the absence of sensory input, unilateral reaching movements remain asymmetric, reflecting control from both hemispheres, but with greater reliance on the contralateral hemisphere, as described by the complementary dominance hypothesis. Most importantly, this study shows that the distinction in control mechanisms between hemispheres is not solely dependent on sensory feedback.

**Summary.** The evidence presented in this section demonstrates that there are distinct behavioural advantages when making movements with either the dominant, right arm or non-dominant, left arm that are characterized under the complementary dominance hypothesis. Dominant arm movements appear to optimally account for intersegmental dynamics under familiar task conditions to minimize trajectory errors and energy expenditure, whereas non-dominant arm movements show an advantage controlling postural limb impedance under unfamiliar or unpredictable task conditions. In the following section we review the

evidence of the complementary dominance hypothesis in unilateral stroke survivors, where neural correlates for lateralized motor functions are identified according to movement deficits resulting from damage to either the left or right cerebral hemisphere.

## Hemispheric lateralization for motor control

Lesion models afforded by studying the movements of stroke survivors have provided substantial information about hemispheric lateralization that reflect the behavioural asymmetries described in the previous sections. Importantly, this research has provided translational and clinical implications in the understanding and treatment of post-stroke motor deficits. In this section, we discuss research that uses stroke survivors as lesion models to better understand hemispheric lateralization for motor control, along with translational studies that utilized the complementary dominance model to remediate post-stroke motor deficits.

**Stroke.** The clinical definition of stroke encompasses any CNS cell death of vascular aetiology (Donkor, 2018; Sacco et al., 2013). Strokes are often classified into one of two categories: (1) ischaemic, such that damage occurs due to obstruction of blood supply (i.e. infarction), and (2) haemorrhagic, such that damage occurs from blood vessel rupture and/or subsequent focal blood collection (Sacco et al., 2013). Additional subtypes may be used but are outside the scope of this review (for a review see Sacco et al., 2013). Approximately 87% of strokes are ischaemic, and this is often classified via computed tomography (CT) scan or magnetic resonance imaging (MRI; Sacco et al., 2013; Tsao et al., 2023). Ischaemic stroke may be unilateral or bilateral, but among unilateral strokes, there is a slightly higher incidence of left hemisphere stroke (~54%) when compared to right hemisphere stroke (~46%; Hedna et al., 2013). The vasculature most prone to ischaemic stroke is the middle cerebral artery (MCA), as this region comprises over half of all strokes (Ng et al., 2007). Other neuroanatomical regions often used to classify stroke location include anterior cerebral artery (ACA), posterior cerebral artery (PCA), cerebellar, brainstem, small-vessel, and those consisting of multiple vascular territories (Ng et al., 2007). In summary, the most prevalent strokes are ischaemic, most often affecting the left MCA.

In the USA, ~85% of stroke victims survive their stroke, but over half of these individuals experience long-term disability as a result (Tsao et al., 2023). This is often due, in part, to lasting motor deficits, among many other comorbid complications (Donkor, 2018; Tsao et al., 2023). Approximately 85% of individuals who have experienced a stroke demonstrate sensorimotor deficits (Nichols-Larsen et al., 2005). These can manifest as muscle paresis, paralysis or contracture, along with impaired coordination (Lang et al., 2013; Sathian, et al., 2011), and have been associated with an inability to return to employment, decreased quality of life and increased caregiver burden (Donkor, 2018; Schindel et al., 2021; Tsao et al., 2023). Increasing the efficacy of post-stroke motor rehabilitation remains an immense, worldwide public health priority (Tsao et al., 2023).

**Ipsi- and contra-lesional motor deficits.** The complementary dominance model has direct implications for individuals with unilateral stroke. According to this model, a unilateral lesion should result in hemisphere-specific motor control deficits in *both* arms, based on the specializations of the lesioned hemisphere. Sensorimotor deficits in the arm contralateral to the brain lesion are common following MCA stroke, including muscle tone deficits, abnormal motor synergies, paresis and weakness (Dewald et al., 1995, 2001; Lang et al., 2013; McPherson & Dewald, 2022; Sathian et al., 2011). However, deficits in the arm ipsilateral to the stroke lesion have been less well recognized, and this arm is most often considered unimpaired in neurological and rehabilitation clinics. Nevertheless, several lines of research have demonstrated the presence of motor deficits in the ipsilesional arm of stroke survivors (Haaland & Harrington, 1989a, 1989b, 1996; Maenza et al., 2020, 2021; Schaefer et al., 2007, 2009a; Winstein & Pohl, 1995). Further, these ipsilesional deficits present differently depending on the hemisphere damaged (Maenza et al., 2020; Sainburg et al., 2016; Schaefer et al., 2007, 2009a). For example, Winstein and Pohl (1995) compared ipsilesional arm reaching performance between individuals with left and right hemisphere stroke as they completed a rapid reciprocal target aiming task. Regardless of the hemisphere affected, stroke participants took longer to complete the task compared to neurologically intact controls. However, individuals with left hemisphere strokes performed the task with longer acceleration phases early in each movement, whereas individuals with right hemisphere strokes performed the task with longer deceleration phases late in each movement. These findings not only demonstrate that the ipsilesional arm exhibits motor deficits but also support the hypothesis that the nature of ipsilesional motor deficits is dependent on the specializations of the lesioned hemisphere.

In a similar line of research, Haaland & Harrington (1989a, 1989b, 1996) had individuals with right and left hemisphere strokes perform a target aiming task with their ipsilesional arm. They showed that individuals with left hemisphere stroke were less accurate in the early, initial aiming of movements during each reach (Haaland & Harrington, 1989b), whereas individuals with

right hemisphere strokes performed with larger errors in stopping on the target (Haaland & Harrington, 1989a). These findings supported the hypothesis that ipsilesional deficits depend on the hemisphere that was damaged by stroke. Specifically, left hemisphere strokes affect the early phase of reaching, whereas right hemisphere strokes affect the decelerative aspects of reaching movements that lead to the ability to stabilize the arm at the end of motion.

Schaefer et al. (2009a) compared ipsilesional upper extremity reaching kinematics and joint torques between right-handed individuals with left and right hemisphere cortical stroke, along with healthy controls. Individuals with left hemisphere damage performed the reaching task with their left arm (example hand path shown in Fig. 6A; blue, left) and showed higher variability in initial reaching direction but relatively preserved final position accuracy (Fig. 6B and C). Further, this variability was higher during movements with higher peak elbow

muscle torque, suggesting deficits in coordination of inter-segmental dynamics. This pattern is consistent with the findings of motor lateralization studies in healthy controls (Bagesteiro & Sainburg, 2002; Sainburg, 2002; Sainburg & Kalakanis, 2000), which concluded that the dominant arm and its respective control system are specialized for the coordination of intersegmental dynamics. Figure 6A (red, right) shows an example hand path for a right hemisphere stroke participant, demonstrating the low movement curvature but impaired final position accuracy. Indeed, individuals with right hemisphere damage had higher errors and increased variability in stopping on the target (Fig. 6B and C), thus demonstrating deficits achieving an accurate final position (Schaefer et al., 2009a). This is also consistent with findings of motor lateralization studies with healthy controls (Bagesteiro & Sainburg, 2003; Sainburg, 2002; Schabowsky et al., 2007), which concluded that the non-dominant arm and its respective

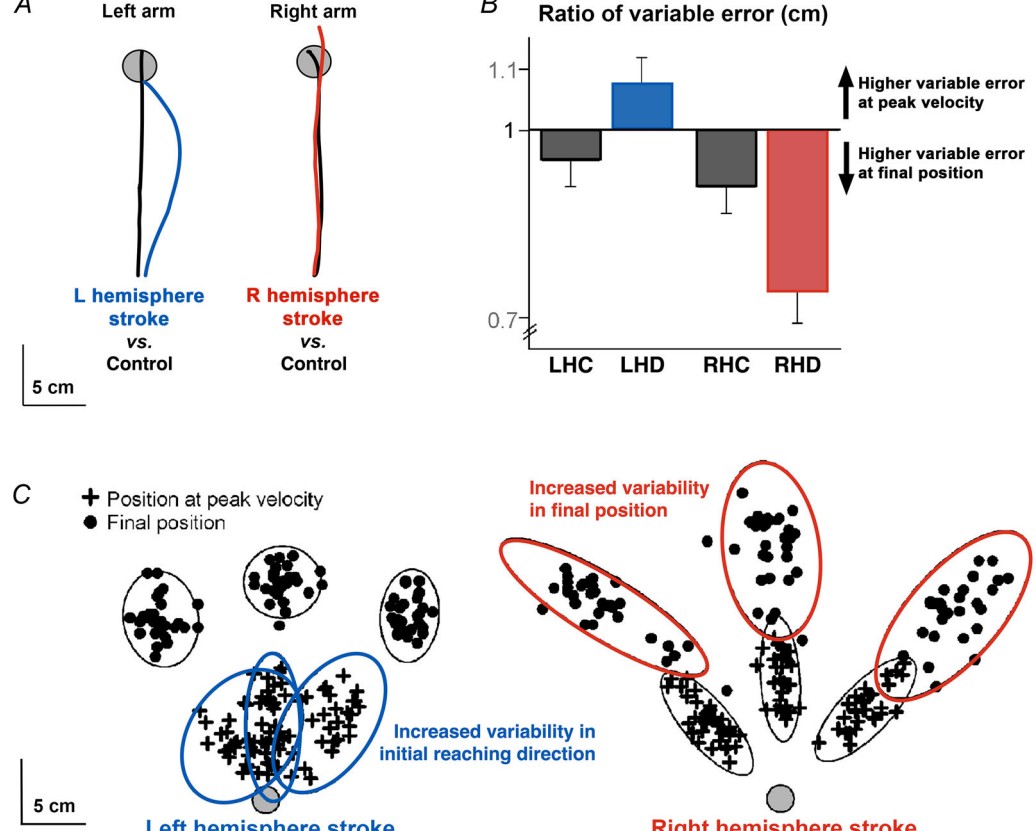

**Figure 6. Deficits in ipsilesional arm reaching performance are dependent on the lesioned hemisphere**
*A*, representative hand trajectories from control participants (black traces), left hemisphere stroke survivors (blue, left) and right hemisphere stroke survivors (red, right). *B*, mean (with 1 standard error bars) ratio of variable error at peak velocity to variable error at final position. L/RHC = healthy controls (HC) using their left (L) or right (R) arms; L/RHD = left (L) or right (R) hemisphere damaged (HD) stroke survivors using their ipsilesional arms. C, hand location at peak velocity (cross symbol) and final position (shaded circle) for individual trials of representative left hemisphere (left, blue) and right hemisphere (right, red) stroke participants. Ellipses are generated from 95% confidence intervals of data from each target.
Figure modified from Schaefer et al. (2009a).

control system are specialized for impedance control. It is important to note that the stroke survivors in this study had unilateral, cortical involvement with minimal subcortical grey matter affected. These studies challenge the idea that ipsilesional deficits can be attributed solely to post-stroke cognitive deficits or slowed, 'one-handed' performance of activities of daily living. In summary, the studies described above support the overarching hypothesis of a bi-hemispheric, complementary model of motor control. Notably, these studies also indicated that the 'non-affected' or 'non-paretic' side in individuals with unilateral stroke should more accurately be referred to as the 'less affected' side.

To gain a better understanding of where in each hemisphere the specializations for impedance and optimal control might reside, Mutha et al. (2011a) quantified motor adaptation processes in the ipsilesional arm of individuals with focal damage to frontal or parietal regions. Twenty unilateral chronic stroke survivors (10 with left hemisphere stroke, 10 with right hemisphere stroke, shown in Fig. 7A), along with 20 healthy controls, completed a point-to-point reaching task in a 2D virtual reality workspace. After a baseline familiarization period, a 30-degree rotation was applied to the cursor representing each participant's fingertip location. Under the complementary dominance hypothesis, unilateral lesions should result in contralesional *and* ipsilesional motor deficits that are dependent on the hemisphere affected. In agreement with this hypothesis, they showed that the participants with left parietal lesions had

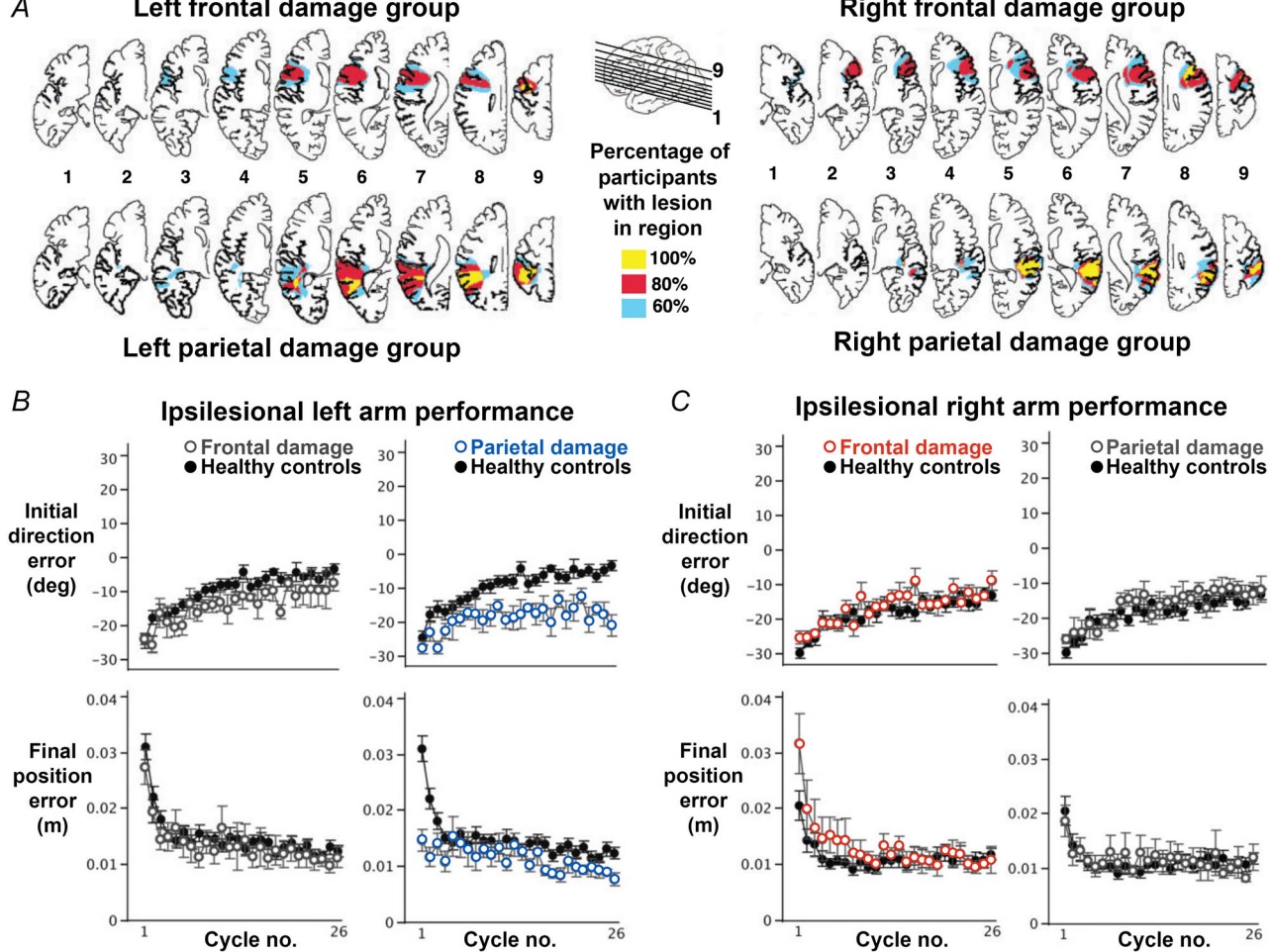

**Figure 7. Neural correlates of lateralized motor behaviour during a visuomotor adaptation task in left and right hemisphere stroke**

*A*, lesion mapping for included participants with frontal or parietal damage affecting either the left or right hemisphere. *B* and *C*, left arm reaching performance of healthy controls and left hemisphere stroke survivors (*B*) and right arm reaching performance of healthy controls and right hemisphere stroke survivors (*C*) over repeated trials under a 30-degree visual perturbation. Each data point represents group means (with 1 standard error bars) of trials binned into eight trial cycles (one to each of the eight different targets). Figure modified from Mutha et al. (2011a).

deficits in adapting initial reaching direction with their ipsilesional arm, as shown in Fig. 7*B* (Mutha et al., 2011a). While initial direction error between the groups did not differ in the first cycle following introduction of the cursor rotation, participants with left parietal lesions had significantly larger initial direction errors in the *last* cycle of the task when compared to the control and right frontal lesion groups. Thus, the participants with right parietal lesions, along with those with left and right frontal lesions, did not differ from healthy control participants in adapting initial reaching direction. The participants with right frontal lesions did, however, show deficits in the final position accuracy of movements made with the novel cursor rotation (Fig. 7*C*). In the first cycle of the task, this group had significantly larger final position errors, while those with left frontal lesions, along with those with left and right parietal lesions, did not demonstrate this deficit when compared to healthy controls. Taken together, these findings suggest that there are distinct, lateralized mechanisms for directional adaptation and for final position achievement. This study also provided support for the complementary dominance hypothesis by identifying specific tertiary processing regions that have an ipsilateral role in a crucial aspect of motor control: motor adaptation. Notably, these regions are often affected by MCA stroke. Thus, an enhanced understanding of lateralized processes has important implications for better prediction and treatment of post-stroke motor deficits.

In a separate study, Mutha et al. (2011b) quantified motor deficits in the *contralesional* (i.e. opposite side) arm of 10 individuals with focal, unilateral parietal lesions, along with 14 healthy controls. Due to the focality of these parietal lesions (five with left parietal damage, five with right parietal damage), sensorimotor cortices were spared, and no observable paresis was present in the contralesional arm of these participants. This allowed for the more precise assessment of parietal cortex contributions to contralesional arm control. All participants performed a similar point-to-point reaching task as discussed above. They showed that healthy control participants, along with those with right parietal damage, adapted to the cursor rotation at similar rates and extents by adjusting initial reaching direction. However, those with left parietal damage continued to reach with significant amounts of initial reaching direction error. While the control and right parietal damage participants showed after-effects immediately following removal of the rotation, the left parietal damage group did not. Taken together, these two studies support the hypothesis that left, but not right, parietal cortices are critical for motor adaptation in *both* arms. Additionally, this study demonstrates that motor deficits in the contralesional arm present differently depending on which hemisphere is lesioned (Mani et al., 2013; Mutha et al., 2011b; Varghese & Winstein, 2020).

In another study of hemisphere-dependent contralesional motor deficits in individuals with larger MCA lesions, Mani et al. (2013) reported a similar pattern of motor deficits: those with right hemisphere stroke performed with higher final position errors in their contralesional arm when compared to the control group. Those with left hemisphere stroke achieved an accurate final position but had deficits in specifying an accurate initial reaching direction towards the target. Thus, their reaching trajectories were more curved, and their initial direction errors were higher and more varied than the control group. These findings support the hypothesis that unilateral stroke results in specific contralesional motor deficits that depend on the hemisphere that was damaged (Mani et al., 2013).

**Ipsilesional deficits and functional independence.** Several lines of research have now shown that ipsilesional deficit severity is directly related to contralesional deficit severity, and these ipsilesional deficits are *functionally limiting* (Chestnut & Haaland, 2008; Hmaied Assadi et al., 2022; Maenza et al., 2020; Poole et al., 2009; Wetter et al., 2005). For example, we quantified contralesional motor capacity using the Upper Extremity Fugl–Meyer (FM), a widely used measure of post-stroke motor impairment (Gladstone et al., 2002), in 110 individuals with chronic unilateral stroke (62 right-hemisphere-damaged, 48 left-hemisphere-damaged; Maenza et al., 2020). In these same individuals, we quantified ipsilesional functional motor impairment using the Jebsen–Taylor Hand Function Test (JHFT). As shown in Fig. 8 (inset), this assessment requires participants to perform a variety of upper extremity tasks mimicking activities of daily living, such as writing, picking up small household objects and simulated feeding. We found that individuals with the most severe contralesional motor deficits (lower FM scores) also had the most severe ipsilesional motor deficits (increased time taken to complete JHFT), and this effect was largest in individuals with left hemisphere damage, who had greater ipsilesional deficits when compared to those with right hemisphere damage (Fig. 8). These findings have substantial clinical implications. Most often, severe contralesional motor deficits prevent functional grasp and release, making this arm an ineffective manipulator. Thus, these individuals must resort to completing all or most activities of daily living with their ipsilesional arm, alone. This line of research suggests that their ipsilesional arm may be a poor primary controller.

Schaefer et al. (2009a) investigated the relationship between specific kinematic aspects of ipsilesional arm performance and scores on the JHFT. Individuals with chronic left or right hemisphere stroke performed targeted reaching movements with their ipsilesional arm. Those

with left hemisphere damage performed with larger errors and variability in initial movement direction, along with larger hand path curvatures throughout each reach. Those with right hemisphere damage had no deficits in initial movement direction but demonstrated longer reaction times and greater errors stopping accurately on the target. When correlating these measures with functional impairment, the researchers found that hand path curvature during these movements strongly predicted ipsilesional JHFT scores for the individuals with *left hemisphere damage only*. This effect was observed despite both groups (left hemisphere damage, right hemisphere damage) having similar JHFT scores. Conversely, reaction time strongly predicted ipsilesional JHFT scores for the individuals with *right hemisphere damage only*. These findings support the hypothesis that functional impairments of ipsilesional deficits can be attributed to different, hemisphere-specific mechanisms. However, in this case, the relationship between reaction time and JHFT scores was not explained by our complementary dominance model, but more probably resulted from perceptual deficits associated with right-hemisphere damage.

**Motor lateralization to inform targeted neuro-rehabilitation.** As previously discussed, ipsilesional motor deficits are most severe in individuals with severe contralesional motor deficits (Maenza et al., 2020). Because of this, it has been suggested that targeted neuro-rehabilitation of the ipsilesional arm in chronic stroke survivors should increase functional independence. We tested this hypothesis in 13 individuals with chronic (>6 months following stroke), unilateral stroke (five with right hemisphere damage, eight with left hemisphere damage; Maenza et al., 2021). These individuals received 3 weeks of ipsilesional arm training (three sessions per week, 1.5 h per session) that was designed to *remediate* the hemisphere-specific deficits produced by their unilateral stroke, followed by 3 weeks of sham training to control for the non-specific effects of therapy visits and interactions (Fig. 9*A*). The ipsilesional arm training sessions included real life dexterous activities and hemisphere-specific

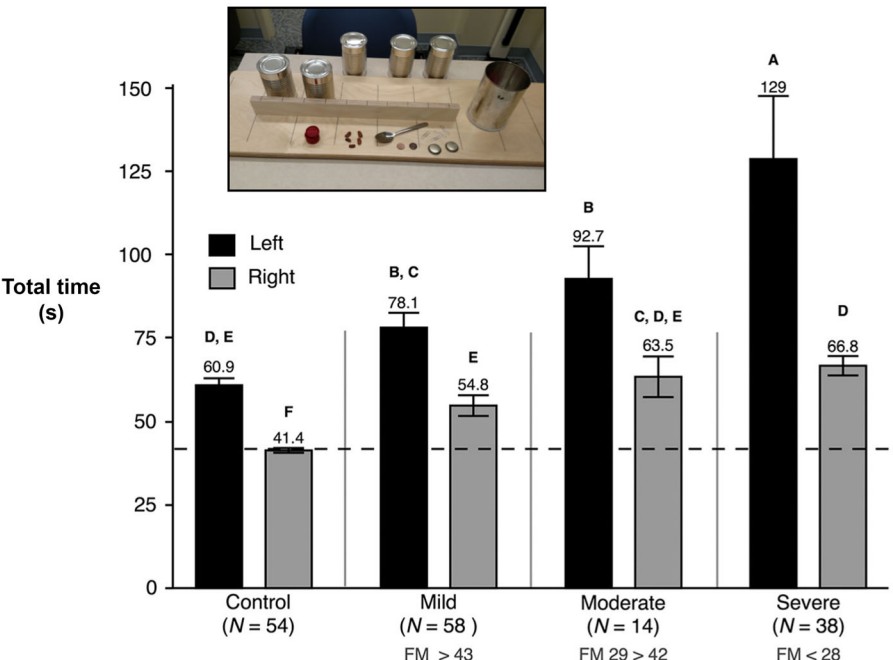

**Figure 8. Motor deficits in the ipsilesional arm of stroke survivors vary with extent of contralesional arm impairment**
Example materials (inset, top) and mean recorded total time in seconds (with 1 standard error bars) on the Jebsen–Taylor Hand Function Test for stroke survivors and neurotypical controls. Stroke survivors were grouped by the severity of their contralesional arm motor impairment, which was measured via Upper Extremity Fugl–Meyer (FM) score. Separate groups of neurotypical controls used either their left or right arm to perform the task, whereas stroke survivors used their ipsilesional arm. Bars not connected by the same letter indicate groups that are significantly different from each other.
Figure modified from Maenza et al. (2020).

virtual reality games. Depending on which hemisphere was damaged, the participants completed an upper extremity 2D virtual reality task that trained either predictive aspects of control (virtual shuffleboard task for individuals with left hemisphere damage; Fig. 9*B*) or feedback-mediated position control (virtual tracing task for individuals with right hemisphere damage; Fig. 9*C*).

Clinical assessment scores were gathered at two baseline time points (Tests 1 and 2, 3 weeks apart) before the intervention protocol, along with at various time points after the targeted intervention protocol (Fig. 9*A*). Despite

being in the chronic phase of stroke, these individuals demonstrated 14.38% higher scores on the self-care portion of the Functional Independence Measure (FIM) and 19.21% decrease in total time required to complete the JHFT. Notably, these scores were maintained at both follow-up testing sessions, as shown in Fig. 9*D* and *E* (Tests 4 and 5, 3 week and 6 week follow-ups, respectively), and did not correspond with any significant functional decline in the contralesional limb, as measured by the Upper Extremity FM score. This pilot study indicated that post-stroke ipsilesional deficits benefit from rehabilitation

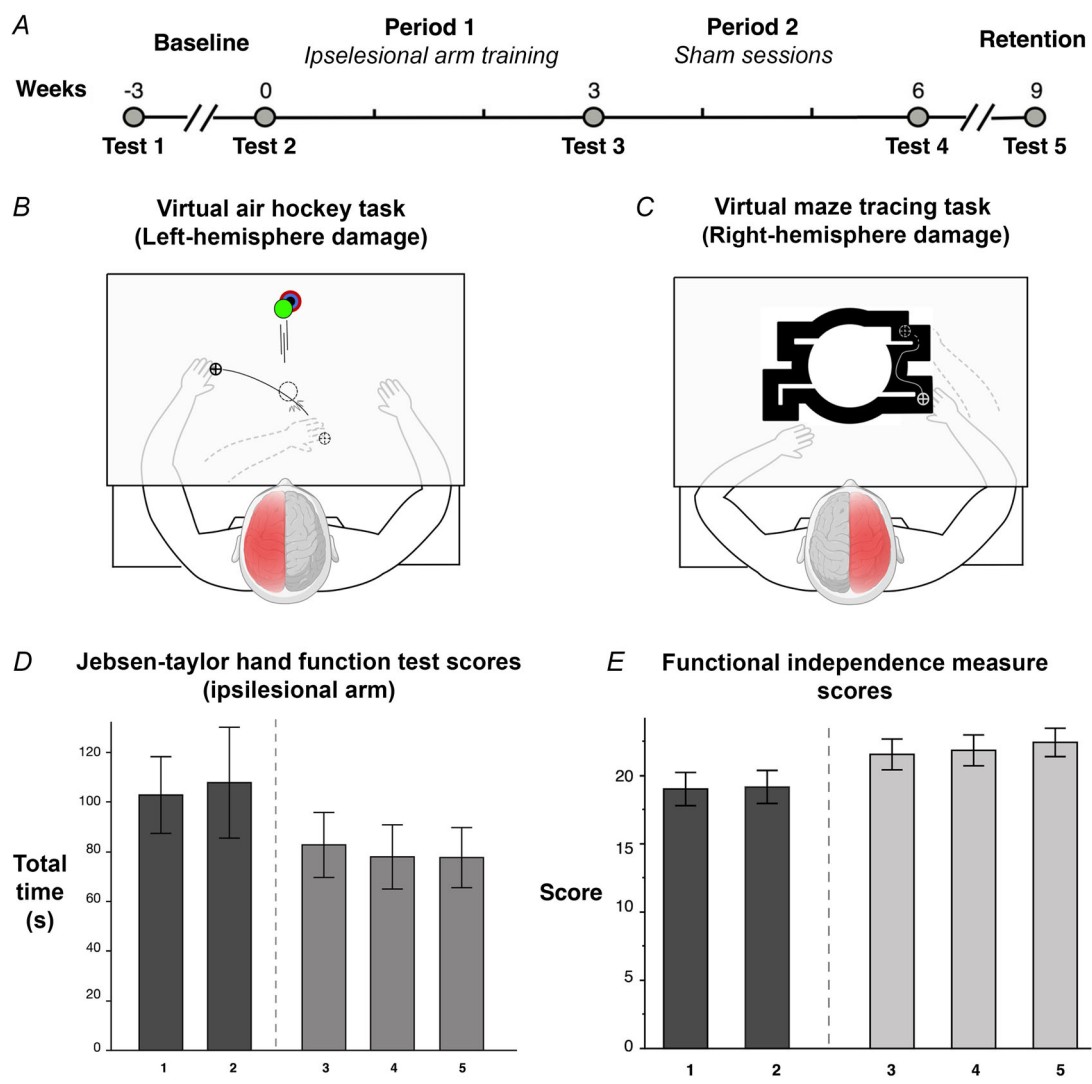

**Figure 9. Pilot study targeting remediation of ipsilesional arm function to improve functional independence**
*A*, study timeline. *B*, ipsilesional arm training task for left hemisphere damaged stroke survivors; participants used their left arm to strike a virtual puck (green circle) towards a target bullseye. *C*, ipsilesional arm training task for right hemisphere damaged stroke survivors; participants used their right arm to trace within the confines (black shaded area) of different shapes. *D*, mean total time to perform the Jebsen–Taylor Hand Function Test; and *E*, mean Functional Independence Measure scores (with 1 standard error bars), recorded at different test points in the study timeline (dashed vertical line indicates when training intervention occurred).
Figure modified from Maenza et al. (2021).

with a hemisphere-specific remediation strategy (i.e. aiming to restore function in the arm), and remediation of these deficits can substantially impact functional independence. Based on these promising pilot findings, we are conducting a two-site, phase II randomized clinical trial in unilateral stroke survivors with severe contralesional upper extremity paresis and functionally limiting ipsilesional deficits (Maenza et al., 2022). This study comprises two groups receiving unilateral arm training: an experimental group that receives 1 h of ipsilesional arm training three times per week for 5 weeks, and a control group that receives an equal amount of conventional therapy on their contralesional arm. The ipsilesional arm training includes virtual reality training targeted at remediating hemisphere-specific deficits similar to the pilot study, followed by real-life dexterity training of their ipsilesional arm focused on manipulating everyday objects. Primary outcome measures will assess ipsilesional arm function (JHFT), contralesional arm impairment (FM) and functional independence (Barthel Index). The Barthel Index is a self-report questionnaire with 10 items related to self-care and mobility, asking respondents to report their level of functional independence on these weighted items. Participants are assessed on ipsilesional and contralesional measures at five time points with the last assessment at 6 months following the end of the training period. We hypothesize performance will improve for the trained arm in both groups, but that ipsilesional arm training will lead to greater improvement in functional independence, presumably through improvements in Barthel Index score items that require dexterous arm use (Maenza et al., 2022).

Additionally, the complementary dominance model may have clinical implications within 'cross-education', or the carry-over of performance to the opposite, untrained limb following unilateral limb training. While a complete discussion on cross-education is outside of the scope of this review (for an in-depth review see Calvert & Carson, 2022), it has been suggested that cross-education could be utilized in post-stroke rehabilitation (Farthing & Zehr, 2014). In other words, rehabilitation of the less-affected limb may result in motor capacity improvements in the more-affected limb. Our laboratory has previously investigated how lateralized specializations within the complementary dominance model may affect the direction and magnitude of cross-education. In neurologically intact young adults, we have shown that carry-over effects following adaptation to a visuomotor rotation are asymmetric and depend on the arm used for initial training. Initial dominant arm training results in carry-over of learned final position accuracy to the non-dominant arm, while initial non-dominant arm training results in carry-over of initial direction accuracy to the dominant arm (Sainburg & Wang, 2002). These findings suggest that the information accessible to the

opposite, untrained limb via cross-education depends on the specialization of that limb's control system. However, it is worth noting that these experiments were conducted in neurologically intact young adult participants. The extent to which these asymmetries might be utilized in rehabilitation to improve contralesional arm motor capacity and functional independence remains unclear. Some rehabilitation strategies, such as constraint-induced movement therapy (CIMT; Wolf et al., 2006), discourage or even restrain ipsilesional arm use, with the assumption being that reduced use of the contralesional arm during ipsilesional arm use may result in deterioration of contralesional arm motor capacity (Taub et al., 2002; Wolf et al., 2002). However, CIMT has been targeted for patients with moderate to mild paresis, who have the ability to grasp and release with the contralesional arm. In contrast, our pilot data suggest that – for chronic stroke populations with severe contralesional paresis who lack the ability to grasp and release with this limb – there is no loss of motor capacity in the contralesional arm following hemisphere-specific ipsilesional arm training (Maenza et al., 2021). In fact, we observed a small, but statistically significant improvement in contralesional arm FM scores. The exact mechanisms underlying this contralesional arm improvement, including whether they might be attributable to cross-education, require further investigation. While this was not a focus of the pilot study referenced above, a recent scoping review (Lim & Madhavan, 2023) and meta-analysis (Smyth et al., 2023) both concluded that cross-education is a promising strategy within rehabilitation to promote contralesional arm function. Thus, additional research is necessary to determine how the complementary dominance model may inform cross-education strategies in stroke rehabilitation.

These clinical trials closed the translational loop from basic studies of motor lateralization to clinical studies of hemispheric lateralization, and finally to clinical intervention. It should be noted that these studies, along with other preliminary studies addressing ipsilesional deficits, were conducted in individuals in the chronic phase of stroke (Maenza et al., 2021, 2022; Pandian et al., 2014). Current physical rehabilitation of acute and subacute stroke remains focused on remediation of the contralesional limb. However, if remediation of the ipsilesional arm in chronic stroke survivors with severe contralesional paresis proves to be beneficial and increases functional independence, future research should assess the generalizability to the acute and subacute phases of rehabilitation.

**Summary.** Sensorimotor deficits are commonly experienced by stroke survivors, which contribute to chronic disability and reduced quality of life. Although

motor impairments of the contralesional arm often receive the most attention during rehabilitation, there is now considerable evidence demonstrating the presence of motor deficits in the ipsilesional arm, despite it often being considered as 'unaffected' by unilateral stroke. Consistent with our complementary dominance hypothesis, the motor deficits that are apparent in this arm following unilateral stroke reflect the specialization of the damaged hemisphere. That is, left hemisphere stroke tends to result in poor coordination of intersegmental dynamics that impairs regulation of the early direction accuracy of movements. By contrast, right hemisphere stroke tends to result in more variable final positions at movement end, indicative of impaired limb impedance control. Moreover, ipsilesional motor deficits vary with the extent of contralesional arm impairment, which leads to the greatest functional deficits for individuals with severe contralesional arm paresis, who must use the ipsilesional arm as the main or sole manipulator. To date, we have conducted two clinical trials to test the efficacy of remediation of the ipsilesional arm with hemisphere-specific training interventions as a means to improve functional independence for chronic stroke survivors. We have reported positive results from a small study of 13 chronic stroke survivors with severe contralesional paresis. Currently, we are concluding a large-scale clinical trial with severely paretic stroke survivors, in which we compare remediation targeting only the contralesional arm to that of targeting only the ipsilesional arm. We predict greater carry over to functional independence for the group with the ipsilesional arm intervention.

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

## Additional information

### Competing interests

The authors report no competing interests.

### Author contributions

R.L.S. conceived of, oversaw and contributed to the editing of all elements of the manuscript. N.M.K. and B.D. played a major role in the writing and editing of all sections. C.M., J.Y., P.R.R.

and T.K. contributed substantially to writing and editing of the manuscript. All authors approved the final version of the manuscript and all persons designated as authors qualify for authorship.

## Funding

This work was supported by NIH grant (2 R01 HD059783-06A1) and Dorothy Foehr Huck and J. Lloyd Huck Distinguished Chair Endowment awarded to R. L. Sainburg.

## Keywords

complementary dominance, motor control, motor lateralization, rehabilitation, stroke

## Supporting information

Additional supporting information can be found online in the Supporting Information section at the end of the HTML view of the article. Supporting information files available:

**Peer Review History**

