## [Peer Review History · The Journal of Physiology]

The Complementary Dominance Hypothesis: A model for remediating the "good" hand in stroke survivors

Nick M Kitchen, Brooke Dexheimer, Jisung Yuk, Candice Maenza, Paul R Ruelos, Taewon Kim, and Robert L Sainburg
DOI: 10.1113/JP285561

Corresponding author(s): Robert Sainburg (rls45@psu.edu)

The following individual(s) involved in review of this submission have agreed to reveal their identity: Warren G Darling (Referee #1); Michael D Ellis (Referee #2)

Review Timeline:

Submission Date:	03-Feb-2024
Editorial Decision:	15-Mar-2024
Revision Received:	12-Apr-2024
Accepted:	25-Apr-2024

Senior Editor: Laura Bennet

Reviewing Editor: Richard Carson

Transaction Report:

Dear Dr Sainburg,

Re: JP-TR-2024-285561 "The Complementary Dominance Hypothesis: A model for remediating the "good" hand in stroke survivors" by Nick M Kitchen, Brooke Dexheimer, Jisung Yuk, Candice Maenza, Paul R Ruelos, Taewon Kim, and Robert L Sainburg

Thank you for submitting your manuscript to The Journal of Physiology. It has been assessed by a Reviewing Editor and by 2 expert referees and we are pleased to tell you that it is acceptable for publication following satisfactory revision.

ABSTRACT FIGURES: Authors may use The Journal's premium BioRender account to create/redraw their Abstract Figures (and any other suitable schematic figure). Information on how to access this account is here: <https://physoc.onlinelibrary.wiley.com/journal/14697793/biorender-access>.

REVISION CHECKLIST: Upload a full Response to Referees file. To create your 'Response to Referees' copy all the reports, including any comments from the Senior and Reviewing Editors, into a Microsoft Word, or similar, file and respond to each point, using font or background colour to distinguish comments and responses and upload as the required file type.

We look forward to receiving your revised submission.

Yours sincerely,

EDITOR COMMENTS

Reviewing Editor:

There is consensus that this is an excellent review that is likely to make an important contribution to the scientific literature and exhibits the potential for clinical impact. By and large the reviewers offer suggestions concerning ways in which the presentation might be refined further. For example, although the figures included with the paper are excellent, some minor modifications (recommended by both reviewers) are likely to be beneficial.

With respect to the initiation of clinical interventions arising from the program of research that is described, it may be worthwhile to discuss convergence with recommendations that have been made recently concerning exploitation of the phenomenon of "cross education" (e.g., Farthing & Zehr, 2014; *Exerc Sport Sci Rev.* 2014 Apr;42(2):70-5). and those that question the mechanistic basis of functional improvements arising from mirror therapy.

Please also see 'Required Items' below.

REFEREE COMMENTS

Referee #1:

This is a clearly written topical review of the scientific evidence for the complementary dominance hypothesis/model of motor lateralization. There is a strong rationale for providing a comprehensive review of previous scientific literature related to the roles of the left and right brain hemispheres in control of unimanual and bimanual movements. I found no major flaws, but there are some issues to consider as described below.

1. There are multiple locations in the text with a sentence such as: "It is still widely believed that each cortical hemisphere exerts control over the contralateral arm only, without influence of the ipsilateral cortices." This is probably true among clinicians (neurologists, PTs, OTs), although I don't know if there is any survey evidence for this, but I think most scientists studying upper limb motor control are aware that the motor areas in the cortex ipsilateral to the moving upper limb are quite active and contribute in some way to controlling that limb in neurologically intact people. Given that this review is most likely to be read primarily by scientists who study upper limb movement control, the sentence should be modified to reflect this.
2. Lines 104-110 - Although activation of ipsilateral hemisphere motor areas is lower during dominant arm movements than during non-dominant arm movements, the point should be emphasized that there are relatively high levels of activation in ipsilateral motor areas (especially premotor areas) during unimanual movements. This has been shown in fMRI studies in humans (e.g., Kansaku et al. 2005 *Neuroimage* 28:69) and in some studies using intracranial monitoring for localization of an epileptogenic focus (e.g., Ganguly et al. 2009 *J Neurosci* 29:12948). Similarly, neuronal recordings in macaques have demonstrated that in secondary motor areas some neurons are exclusively related to ipsilateral hand movements (e.g., Tanji et al. 1988 *J Neurophys* 60:325). Adding some discussion to show that ipsilateral motor areas are active and contributing to control of ipsilateral arm movements would be appropriate.
3. Fig. 2,4,5 - Calibration bars for the movement trajectories should be added to these figures. Figure 6 contains calibration bars for motion in the anterior and right/left directions but figures 2 and 4 do not have these and Fig. 5 only shows a calibration bar for motion in the right/left direction.
4. Fig. 4, lines 257-259 and 261-263 - Differences in linearity of right and left arm movements appear to be rather small in Fig. 4D. The straight line distance for these movements was 20 cm. According to Fig. 4D the left arm movements averaged about 7-9% or 1.4-1.8 cm deviation from a straight line and the right arm movements averaged about 5-7.5% or 1-1.5 cm deviation from a straight line. The average difference in deviation from linearity was only about 3 mm. Similarly, differences between left and right hand displacements when stabilizing against the spring load were only about 2 mm according to Fig. 4G. Although these differences were statistically significant, the physiological significance in relation to slicing and stabilizing a loaf of bread is questionable. The text should indicate this is relatively weaker evidence for the model in comparison to some of the other evidence presented.
5. Lines 483-488 - Fig. 7B does not show initial direction errors. Although it could be argued that hand path curvature is related to initial direction errors, it would be simpler to use a figure of initial direction errors (as in the Mutha et al. 2011 paper) to make this point (which would also be consistent with the previous discussion and Fig. 6).

6. Lines 488-491 - According to fig. 7C the differences in final position error in the first cycle were about 0.012 cm. I wonder if this is an error on the scale, which perhaps should be m instead of cm (i.e., patients final position errors were probably about 1 cm larger than those of controls). If the scale is accurate, a difference of 0.012 cm (or 0.12 mm) in final position errors between neurologically intact controls and stroke patients would certainly not be considered physiologically significant and would not be strong evidence for the model.

Referee #2:

JP-TR-2024-285561

The Complementary Dominance Hypothesis: A model for remediating the "good" hand in stroke survivors.

The authors have produced a well-written and concise review of the complementary dominance hypothesis and its current translation from basic to preclinical/clinical research. The authors have provided excellent organization to decades of sophisticated and scientifically rigorous motor control research. This work will serve as a resource and reference for future work and therefore will have great impact on the field. After carefully reviewing the manuscript, I only have a few minor edits/suggestions and one "area of opportunity" for further discussion. The suggestions are itemized below.

Minor edits/suggestions:

Ln 185. "non-dominant arm" is duplicated in this sentence.

Ln 487. It would help to modify Figure 7 to only emphasize the B&C panels where a significant effect was observed. For example, right frontal lesions appear significant in 7C (in initial reaching direction) because of the color highlight. Similarly, for left parietal lesions (final position accuracy). Perhaps only highlight the two of eight panels that were significant.

Ln 505. May help to place "contralesional" [arm control] in italics to emphasize/restate.

Sec. 3d.

-With functional independence being measured by Barthel Index, are there specific items that the authors expect to improve?

Area of opportunity:

-It would strengthen the section to also include a discussion of what is hypothesized to occur on the contralesional arm following ipsilesional arm training. Is carry-over expected considering the discussed (Sec. 3b) evidence that the left parietal cortex is associated with contributions of control to both arms (perhaps right frontal cortex exerts bilateral control as well)? Can increased utilization of contralesional hemisphere (during less-affected arm training) impact the degree to which the more-affected arm is driven via contralesional corticoreticulospinal pathways?

REQUIRED ITEMS

- Please include an Abstract Figure file, as well as the Figure Legend text within the main article file. The Abstract Figure is a piece of artwork designed to give readers an immediate understanding of the Review Article and should summarise the main conclusions. If possible, the image should be easily 'readable' from left to right or top to bottom. It should show the physiological relevance of the Review so readers can assess the importance and content of the article. Abstract Figures should not merely recapitulate other figures in the Review. Please try to keep the diagram as simple as possible and without superfluous information that may distract from the main conclusion of the Review. Abstract Figures must be provided by authors no later than the revised manuscript stage and should be uploaded as a separate file during online submission labelled as File Type 'Abstract Figure'. Please ensure that you include the figure legend in the main article file. All Abstract Figures will be sent to a professional illustrator for redrawing and you may be asked to approve the redrawn figure before your paper is accepted.

- Your MS must include a complete "Additional information section" with the following 4 headings and content:

Competing Interests: A statement regarding competing interests. If there are no competing interests, a statement to this effect must be included. All authors should disclose any conflict of interest in accordance with journal policy.

Author contributions: Each author should take responsibility for a particular section of the study and have contributed to writing the paper. Acquisition of funding, administrative support or the collection of data alone does not justify authorship; these contributions to the study should be listed in the Acknowledgements. Additional information such as 'X and Y have contributed equally to this work' may be added as a footnote on the title page.

It must be stated that all authors approved the final version of the manuscript and that all persons designated as authors qualify for authorship, and all those who qualify for authorship are listed.

Funding: Authors must indicate all sources of funding, including grant numbers. If authors have not received funding, this must be stated.

It is the responsibility of authors funded by RCUK to adhere to their policy regarding funding sources and underlying research material. The policy requires funding information to be included within the acknowledgement section of a paper. Guidance on how to acknowledge funding information is provided by the Research Information Network. The policy also requires all research papers, if applicable, to include a statement on how any underlying research materials, such as data, samples or models, can be accessed. However, the policy does not require that the data must be made open. If there are considered to be good or compelling reasons to protect access to the data, for example commercial confidentiality or legitimate sensitivities around data derived from potentially identifiable human participants, these should be included in the statement.

Acknowledgements: Acknowledgements should be the minimum consistent with courtesy. The wording of acknowledgements of scientific assistance or advice must have been seen and approved by the persons concerned. This section should not include details of funding.

- Please upload separate high quality figure files via the submission form.

- Author profile(s) must be uploaded via the submission form. Authors should submit a short biography (no more than 100 words for one author or 150 words in total for two authors) and a portrait photograph of the two leading authors on the paper. These should be uploaded and clearly labelled together in a Word document with the revised version of the manuscript. Any standard image format for the photograph is acceptable, but the resolution should be at least 300 DPI and preferably more. A group photograph of all authors is also acceptable, providing the biography for the whole group does not exceed 150 words.

- It is the authors' responsibility to obtain any necessary permissions to reproduce previously published material and to list these within the main article file. For information, please see: https://jp.msubmit.net/cgi-bin/main.plex?form_type=display_requirements#permissions.

END OF COMMENTS

Confidential Review

03-Feb-2024

Reviewing Editor:

1. With respect to the initiation of clinical interventions arising from the program of research that is described, it may be worthwhile to discuss convergence with recommendations that have been made recently concerning exploitation of the phenomenon of "cross education" (e.g., Farthing & Zehr, 2014; Exerc Sport Sci Rev. 2014 Apr;42(2):70-5). and those that question the mechanistic basis of functional improvements arising from mirror therapy.

Referee #1:

1. There are multiple locations in the text with a sentence such as: "It is still widely believed that each cortical hemisphere exerts control over the contralateral arm only, without influence of the ipsilateral cortices." This is probably true among clinicians (neurologists, PTs, OTs), although I don't know if there is any survey evidence for this, but I think most scientists studying upper limb motor control are aware that the motor areas in the cortex ipsilateral to the moving upper limb are quite active and contribute in some way to controlling that limb in neurologically intact people. Given that this review is most likely to be read primarily by scientists who study upper limb movement control, the sentence should be modified to reflect this.
 - Thank you for highlighting this. We agree with the reviewer that most of the target audience for this manuscript will indeed be aware that ipsilateral cortex is actively involved in upper limb movement control. However, in this early overview section of the manuscript, we wanted to highlight that this is still often overlooked or neglected in studies of human motor control, which indirectly lends support to global dominance hypotheses of handedness.
 - We do, however, recognize that the wording was too strong and have edited the sentence with softer language to reflect this perspective more accurately (Figure 1 caption and in lines 146-153). Please also see the response to the reviewer's comment below for further discussion.
2. Lines 104-110 - Although activation of ipsilateral hemisphere motor areas is lower during dominant arm movements than during non-dominant arm movements, the point should be emphasized that there are relatively high levels of activation in ipsilateral motor areas (especially premotor areas) during unimanual movements. This has been shown in fMRI studies in humans (e.g., Kansaku et al. 2005 Neuroimage 28:69) and in some studies using intracranial monitoring for localization of an epileptogenic focus (e.g., Ganguly et al. 2009 J Neurosci 29:12948). Similarly, neuronal recordings in macaques have demonstrated that in secondary motor areas some neurons are exclusively related to ipsilateral hand movements (e.g., Tanji et al. 1988 J Neurophys 60:325). Adding some discussion to show that ipsilateral motor areas are active and contributing to control of ipsilateral arm movements would be appropriate.

- As noted above in response to comment (1), we agree with the reviewer that there is ample evidence to indicate functional relevance of ipsilateral cortex activity during unilateral arm movement.
 - The purpose of this early overview section of the review was to provide some alternative views to our own hypothesis of motor lateralization and handedness, hence we had attempted to highlight some of the neurophysiological data that supports these alternative views.
 - In light of the reviewer comments, we acknowledge this may not have been sufficiently balanced with evidence to the contrary, so have now made edits with some of the suggested references to address this on lines 146-153.
3. Fig. 2,4,5 - Calibration bars for the movement trajectories should be added to these figures. Figure 6 contains calibration bars for motion in the anterior and right/left directions but figures 2 and 4 do not have these and Fig. 5 only shows a calibration bar for motion in the right/left direction.
- These have now been added to the figures.
4. Fig. 4, lines 257-259 and 261-263 - Differences in linearity of right and left arm movements appear to be rather small in Fig. 4D. The straight line distance for these movements was 20 cm. According to Fig. 4D the left arm movements averaged about 7-9% or 1.4-1.8 cm deviation from a straight line and the right arm movements averaged about 5-7.5% or 1-1.5 cm deviation from a straight line. The average difference in deviation from linearity was only about 3 mm. Similarly, differences between left and right hand displacements when stabilizing against the spring load were only about 2 mm according to Fig. 4G. Although these differences were statistically significant, the physiological significance in relation to slicing and stabilizing a loaf of bread is questionable. The text should indicate this is relatively weaker evidence for the model in comparison to some of the other evidence presented.
- With respect to Figure 4D and the deviation from linearity data for the reaching hand, the values presented are in arbitrary units (“a.u.”) since the calculation for this measure is detailed in the Woytowicz et al. 2018 paper as follows...
 - *“Hand-path deviation from linearity was defined as the minor axis of the path divided by the major axis of the path. The major axis was defined as the longest distance between any two points parallel to the hand path, and the minor axis was defined as the longest distance between any two points perpendicular to the major axis.”*
 - We have now added a clarification in the figure caption to denote this abbreviated unit of measurement.
 - Regarding the displacement of the stabilizing hand in Figure 4G, we have now made additional comments in lines 313-317 to outline that while statistically significant the

magnitude of interlimb difference in displacement of the stabilizing hand is relatively small.

5. Lines 483-488 - Fig. 7B does not show initial direction errors. Although it could be argued that hand path curvature is related to initial direction errors, it would be simpler to use a figure of initial direction errors (as in the Mutha et al. 2011 paper) to make this point (which would also be consistent with the previous discussion and Fig. 6).
 - We appreciate the reviewer's suggestion and agree that this would make for a more cohesive figure when considering the previous figures. We have modified Figure 7 accordingly.
6. Lines 488-491 - According to fig. 7C the differences in final position error in the first cycle were about 0.012 cm. I wonder if this is an error on the scale, which perhaps should be m instead of cm (i.e., patients final position errors were probably about 1 cm larger than those of controls). If the scale is accurate, a difference of 0.012 cm (or 0.12 mm) in final position errors between neurologically intact controls and stroke patients would certainly not be considered physiologically significant and would not be strong evidence for the model.
 - Thank you for bringing this error to our attention. Based on the reviewer's concerns, we have revisited this dataset and analysis stream. We can confirm that the figure scale is, indeed, incorrectly labeled not only in the current manuscript, but also in the original published study (Mutha et al., 2011). We have contacted the original authors of this paper and initiated a correction for the figures. We have also corrected Figure 7 of the present review accordingly. Again, we appreciate the reviewer bringing this to our attention.

Referee #2:

1. Ln 185. "non-dominant arm" is duplicated in this sentence.
 - Thank you, the has been amended.
2. Ln 487. It would help to modify Figure 7 to only emphasize the B&C panels where a significant effect was observed. For example, right frontal lesions appear significant in 7C (in initial reaching direction) because of the color highlight. Similarly, for left parietal lesions (final position accuracy). Perhaps only highlight the two of eight panels that were significant.
 - With respect to Figure 7B-C, we have opted for this specific pattern of color highlighting to not only emphasize statistically significant differences, but to also highlight where typical performance was observed (for example, no difference in initial direction errors between the right frontal lesion group and control group). We believe this more clearly emphasizes the opposing pattern of deficits we observed within the different lesion groups. Thus, respectfully, we have opted to leave the color highlight for emphasis but

have added explicit statements regarding statistical significance within Section 3B for increased clarity (Lines 532-535 and 539-540).

3. Ln 505. May help to place "contralesional" [arm control] in italics to emphasize/restate.
 - We agree with the reviewer that additional emphasis would be beneficial and have implemented this in the referenced sentence.
4. Sec. 3d. - With functional independence being measured by Barthel Index, are there specific items that the authors expect to improve?
 - Thank you for your question. We have added additional commentary regarding this to Section 3D, Lines 657-660 and 664-665 of the manuscript. In brief, the ipsilesional arm training intervention aimed to remediate function with a combination of real-life dexterous activities and hemisphere-specific virtual reality training sessions. As such, we presume that improvements observed in Barthel Index scores will likely be in self-care items that rely heavily on dexterous arm use (i.e. dressing, grooming, etc.).
5. It would strengthen the section to also include a discussion of what is hypothesized to occur on the contralesional arm following ipsilesional arm training. Is carry-over expected considering the discussed (Sec. 3b) evidence that the left parietal cortex is associated with contributions of control to both arms (perhaps right frontal cortex exerts bilateral control as well)? Can increased utilization of contralesional hemisphere (during less-affected arm training) impact the degree to which the more-affected arm is driven via contralesional corticoreticulospinal pathways?
 - Thank you – we agree with the reviewer's suggestion. We have added additional commentary within Section 3B surrounding carry-over effects resulting from cross-education (Lines 667-700). However, we are unable to comment on specific pathways that may drive these carry-over effects. We do agree with the reviewer that an enhanced understanding of the neuroanatomical correlates for the complementary dominance model could further drive translational and clinical implications within stroke rehabilitation.

Dear Dr Sainburg,

Re: JP-TR-2024-285561R1 "The Complementary Dominance Hypothesis: A model for remediating the "good" hand in stroke survivors" by Nick M Kitchen, Brooke Dexheimer, Jisung Yuk, Candice Maenza, Paul R Ruelos, Taewon Kim, and Robert L Sainburg

We are pleased to tell you that your paper has been accepted for publication in The Journal of Physiology.

Authors should note that it is too late at this point to offer corrections prior to proofing. Major corrections at proof stage, such as changes to figures, will be referred to the Editors for approval before they can be incorporated. Only minor changes, such as to style and consistency, should be made at proof stage. Changes that need to be made after proof stage will usually require a formal correction notice.

Yours sincerely,

Laura Bennet
Senior Editor
The Journal of Physiology

P.S. - You can help your research get the attention it deserves! Check out Wiley's free Promotion Guide for best-practice recommendations for promoting your work at www.wileyauthors.com/eeo/guide. You can learn more about Wiley Editing Services which offers professional video, design, and writing services to create shareable video abstracts, infographics, conference posters, lay summaries, and research news stories for your research at www.wileyauthors.com/eeo/promotion.

IMPORTANT NOTICE ABOUT OPEN ACCESS: To assist authors whose funding agencies mandate public access to published research findings sooner than 12 months after publication, The Journal of Physiology allows authors to pay an Open Access (OA) fee to have their papers made freely available immediately on publication.

You can check if your funder or institution has a Wiley Open Access Account here: <https://authorservices.wiley.com/author-resources/Journal-Authors/licensing-and-open-access/open-access/author-compliance-tool.html>.

EDITOR COMMENTS

Reviewing Editor:

The authors have provided a comprehensive and satisfactory response to all issues highlighted in the initial reviews.

REFeree COMMENTS

Referee #1:

The authors have modified the manuscript appropriately. I now recommend acceptance of the manuscript for publication.

Referee #2:

The authors have sufficiently addressed the issues identified in the initial review.

1st Confidential Review

12-Apr-2024